# Experimental Study and Mechanism Analysis of Preparation of α-Calcium Sulfate Hemihydrate from FGD Gypsum with Dynamic Method

**DOI:** 10.3390/ma15093382

**Published:** 2022-05-09

**Authors:** Ying Li, Wen Ni, Pengxuan Duan, Siqi Zhang, Jiajia Wang

**Affiliations:** 1School of Civil and Resource Engineering, University of Science and Technology Beijing, No. 30 Xueyuan Road, Haidian District, Beijing 100083, China; niwen@ces.ustb.edu.cn (W.N.); zsq2017@ustb.edu.cn (S.Z.); 13811261093@163.com (J.W.); 2School of Materials Science and Engineering, Guilin University of Technology, Jian’gan Road 12#, Guilin 541004, China; duanpengxuan@126.com

**Keywords:** α-calcium sulfate hemihydrate, dynamic method, unit cell, mechanism

## Abstract

Flue-gas desulphurization (FGD) gypsum is a highly prevalent industrial by-product worldwide, which can be an excellent alternative to natural gypsum due to its high content of CaSO_4_·2H_2_O. The preparation of α-calcium sulfate hemihydrate is a high-value pathway for the efficient use of FGD gypsum. Here, a dynamic method, or an improved autoclaved process, was used to produce α-calcium sulfate hemihydrate from FGD gypsum. In this process, the attachment water of the mixture of FGD gypsum and crystal modifiers was approximately 18%, and the pH value was approximately 6.0. The mixture did not need to be pressed into bricks or made into slurry, and it was directly sent into the autoclave reactor for reaction. It was successfully applied to the practical production and application of FGD gypsum, citric acid gypsum and phosphogypsum. In this work, the compositions and morphology of the product at different stages of the reaction were examined and compared. In particular, single-crystal diffraction was used to produce the crystal structure of CaSO_4_·0.5H_2_O, and the results were as follows: *a* = 13.550(3); *b* = 13.855(3); *c* = 12.658(3); β = 117.79(3)°; space group *C*2. The preferential growth along the *c*-axis and the interaction mechanism between the carboxylate groups and the crystal were discussed throughout the analysis of the crystal structure.

## 1. Introduction

Flue-gas desulphurization (FGD) gypsum is an industrial by-product from heating plants using wet desulfurization technology [1,2,3], whose major component is dihydrate gypsum (CaSO_4_·2H_2_O, DH). FGD gypsum is an important alternative to natural gypsum due to its high purity. Currently, FGD gypsum is mainly applied to produce cement retarder [4] and calcined gypsum [5].

Hemihydrate gypsum (CaSO_4_·0.5H_2_O, HH) is a kind of air-hardening cementitious material, which is widely applied in building and decoration materials, molding, special binder systems, precision casting, etc. [6,7]. Depending on different preparation conditions [8,9], it can obtain two forms of hemihydrate gypsum, α- and β- hemihydrate gypsum, which are denoted as α-HH and β-HH, respectively. Many scholars believe that the cell parameters of α-HH and β-HH are identical since no obvious differences between the two forms have been concluded by modern testing and analysis techniques [8,10]. However, their physical properties are quite different, and α-HH can reach a much higher mechanical strength and better working performance than β-HH due to differences in crystallinity [8,9]. Therefore, the preparation of α-HH is one of the most profitable approaches of FGD gypsum treatment [11].

In general, there are two types of industrial processes for producing α-calcium sulfate hemihydrate (α-HH), which are the hydrothermal method under pressure [12,13] and the conventional autoclaved process [14,15]. The conventional autoclaved process is simple, with a lower mechanical strength, whereas the product prepared using the hydrothermal method is well-crystallized and has a higher cost. In fact, there is another method, called a salt solution [16,17] or an alcohol–water solution [18,19] under atmospheric pressure, which is characterized by mild reaction conditions. However, the industrial application of this method is rarely reported due to equipment corrosion, product instability and certain water treatment. Applying a dynamic method is an innovative way to integrate the advantages of the conventional autoclaved process and the hydrothermal method. In the process, the reactants remain in a loose solid state with an attachment water content of approximately 18%, and is dynamic in the reactor, so as to be heated up quickly and homogeneously. Moreover, the excess attachment water helps the crystals grow better, and little water treatment is required. Thus, the dynamic method has the characteristics of being a simple process, having a large capacity, high product quality and low cost, which makes it suitable for the comprehensive utilization of a large amount of industrial by-product waste, such as FGD gypsum, phosphogypsum and citric acid gypsum.

The physical properties of α-HH are closely associated with crystal morphologies, which vary from a columnar to needle-like shape [3,20,21,22]. The needle-like crystals with a high aspect ratio (length to diameter) need a higher water–gypsum weight ratio of standard consistency, and this results in a lower strength, even lower than β-HH. In contrast, the columnar crystals with a low aspect ratio of approximately 1 have a much higher strength due to their much lower water demand. In order to control the morphology of α-HH crystals, many studies have been carried out by scholars through adding different crystal modifiers into the preparation process. Zürz et al. [23] found that this could obviously change the morphology of α-HH crystals by adding a small quantity of carboxylic acids into the salt solution. Panpa and Jinawath [24] took succinic acid or sodium succinate as crystal modifiers in the preparation of α-HH from natural gypsum or FGD gypsum. Yue et al. [10] systematically examined the effects of various modifiers and found that the aspect ratio of the crystals could be effectively reduced to ~1 by using inorganic salt and organic acid together. Tan et al. [25] proved that the relative growth rates of different crystal orientations were changed due to the selective adsorption of maleic acid on different crystal faces. Zhang et al. [26] discussed the growth habits of α-HH crystals in pure water and aqueous solutions containing sulfate acid (salts) or organic acid medium, and they found that the medium had a great influence on the crystal growth. Inspired by these results, potassium sodium tartrate and aluminum sulfate were used as crystal modifiers. Through the dynamic method, the α-HH crystals with a short columnar shape were obtained from FGD gypsum, and the 2 h flexural strength was ~7.0 MPa, whereas the oven-dry compressive strength was ~60 MPa.

In this work, the growth process and crystal structure of α-HH were revealed by using test methods such as powder diffraction, scanning electron microscopy, and single-crystal diffraction. Finally, the mechanism of the morphology transformation of α-HH crystals was discussed.

## 2. Materials and Methods

### 2.1. Materials

Potassium sodium tartrate, aluminum sulfate, sulfuric acid and sodium hydroxide were all purchased from Sinopharm Chemical Reagent CO., Ltd., Shanghai, China. The chemical component of the FGD gypsum was collected from Beijing Guohua Electric Power Corporation (Beijing, China), as shown in Table 1. The XRD pattern of FGD gypsum is provided in Figure 1, revealing DH as the main component. Figure 2 shows the morphology of the FGD gypsum crystals, which is a thick plate shape and obviously different from α-HH crystals. 

### 2.2. Experimental Procedures

Potassium sodium tartrate and aluminum sulfate were first dissolved in the deionized water according to a certain ratio. Then, 5 kg FGD gypsum was mixed with the solution, and the pH value of the mixture was adjusted to approximately 6.0 through the addition of sulfuric acid or sodium hydroxide. Subsequently, the mixture with an attachment water content of approximately 18% was added into the autoclave reactor, heated up to the temperature of 150 °C and left for 30 min. During the reaction process, the mixture was stirred with an impeller at a constant rate of 10 rpm in the reaction stage and 20 rpm in the drying stage.

The samples used for the XRD and SEM tests were withdrawn by terminating the reaction at different stages, washed quickly with anhydrous ethanol three times and then dried at 80 °C in an oven.

### 2.3. Characterization

The physical performances were assessed according to JC/T 2038-2010(China standard for α-high strength gypsum). 

The composition of the samples at different reaction stages was determined through an X-ray diffractometer (XRD, Ultima-IV, Rigaku Inc., Tokyo, Japan) using Cu Kα radiation with a scanning rate of 10°/min and a 2θ range from 5° to 75°.

The crystal morphology was examined using a scanning electron microscope (SEM, S-3400N, Hitachi, Tokyo, Japan). 

The adsorption of the modifiers on the crystal surfaces was analyzed using Fourier transform infrared spectroscopy (Nicolet470, ThermoFisher, Maltham, MA, USA) at a resolution of 4 cm^−1^, with a wavenumber range of 400–4000 cm^−1^.

### 2.4. Single-Crystal Diffraction Experiment for the Crystal Structure 

Single-crystal diffraction was carried out in this work in order to determine the crystal structure and cell parameters of α-HH, which required us to choose a spherical or granular single crystal that was as perfect as possible. However, it is difficult to culture perfect crystals of α-HH, since there are many factors affecting crystal growth, and the crystals are usually small and twinning. The hydrothermal method was used to obtain large and well-crystallized single crystal. The procedure was as follows: A certain amount of deionized water was added into the autoclave reactor, together with the mixture of FGD gypsum, potassium sodium tartrate, aluminum sulfate and sodium hydroxide in specific proportions, and then heated up to the temperature of 150 °C for 30 min. Importantly, the heating rate was low, set as 0.3 °C/min. The stirring rate was 60 rpm, which was higher than that of the dynamic method in order to avoid twin crystals as much as possible. The sample was withdrawn immediately when the reaction was finished, and then quickly washed with hot water at 90 °C, as well as with anhydrous ethanol three times. Finally, the sample was dried in an oven. The prepared crystals were measured using a microscope. A crystal with a length of over 0.08 mm was selected for the single-crystal diffraction test using an X-ray single crystal diffractometer (Saturn 724+, Rigaku Inc., Tokyo, Japan). Usually, crystals would be selected several times, and the data with the strongest diffraction were considered as the final result.

## 3. Results

### 3.1. The Physical Performances

The performance results from the testing of the prepared α-HH plaster according to JC/2038-2010 (China standard) are shown in Table 2. It can be seen that the α-HH plaster has a good mechanical performance and broad application prospects.

### 3.2. Components and Morphology of the Samples at Different Reaction Stages

On the basis of previous research work [28], Figure 3 illustrates the XRD patterns of the samples at different stages of the reaction. As shown in Figure 3, the HH phase began to appear as the temperature reached 140 °C, and there was no characteristic diffraction peak of DH when the temperature rose to 150 °C, indicating the end of DH dehydration. Over time, the characteristic diffraction peaks of HH enhanced, which meant that the crystals gradually improved [29]. Additionally, anhydrite gypsum (CaSO_4_,AH) was not detected in the XPD test.

The crystal morphology of the sample withdrawn at 140 °C is shown in Figure 4a. It was observed that most of the crystals were still DH. In association with the XRD pattern, this confirmed that the major component of the sample was the DH phase accompanied, to a minor extent, by the HH phase. The morphologies of samples withdrawn at 150 °C and 150 °C for 10 min are shown in Figure 4b,c, respectively. The DH crystals are not shown in the pictures. In accordance with the XRD patterns, the main component of the samples was the HH phase. However, the crystal growth had not finished, especially along the *c*-axis direction. The morphology of the sample withdrawn at 150 °C for 30 min is shown in Figure 4d. The fiber shape disappeared and the top facet became flat, indicating that the crystal growth had finished and the synthesis reaction of HH had been completed.

### 3.3. Dehydration of FGD Gypsum with Different Contents of Attachment Water 

In this experiment, DH with 15 wt% and 0 of attachment water was first mixed with identical crystal modifiers, and then placed in a saturated steam environment to prepare the α-HH. Crystals with two different types of morphology were obtained (Figure 5). The crystals in Figure 5a were recrystallized into a hexagonal column shape, whereas the crystals in Figure 5b were cleaved into lamellae, indicating that the crystal water escaped directly from the surfaces of the DH crystals, which did not develop into the morphology of α-HH crystals, while the excess attachment water would help the crystal growth.

### 3.4. FTIR Results of α-HH with Different Modifier Dosages 

FTIR spectroscopy is one important method to study the interaction mechanism between chemicals and the material surfaces [18,30,31,32]. The corresponding characteristic peaks will rise or show displacements after the addition of organic acid or organic acid salt, which proves that the organic acid ions have been adsorbed on α-HH crystal surfaces. The FTIR spectra of α-HH prepared with different dosages (0, 0.15 wt% and 0.28 wt%) of potassium sodium tartrate is shown in Figure 6. As illustrated in Figure 6a, the peaks at 3611 cm^−1^, 3558 cm^−1^ and 1620 cm^−1^ can be attributed to the O-H vibration of the crystal water molecules [10,33]. The double peaks at 1153 cm^−1^ and 1095 cm^−1^ are associated with the stretching vibration of υ3 SO_4_^2−^. The peak at 1008 cm^−1^ is related to υ1 SO_4_^2−^ stretching. The peaks of 663 cm^−1^ and 601 cm^−1^ correspond to υ4 SO_4_^2−^ stretching [20,34]. 

In order to obtain more detailed information on the interaction between potassium sodium tartrate and the α-HH crystals, FTIR spectra was enlarged from 3250 cm^−1^ to 2500 cm^−1^ (Figure 6b). There are three obvious adsorption peaks in all of the curves. Generally, the peak of 3211 cm^−1^ is related to the stretching of O-H in the organic substance [35], and the peaks of 2925 cm^−1^ and 2850 cm^−1^ can be assigned to the asymmetric and symmetric stretching vibrations of methylene(−CH_2_−), respectively [18,36], which implicates that there are certain organics in the samples. This result could be explained by the fact that the FGD gypsum material might contain a small amount of organic substance [18,31]. However, as the dosage of potassium sodium tartrate increased to 0.28 wt%, these stretching vibrations increased in intensity obviously, which might confirm the mutual reaction of the organic acid salt and α-HH crystal [37,38]. Nevertheless, according to the result of single crystal diffraction, neither carbon nor aluminum was found in the interior or edge of the crystal, indicating that the modifiers were merely adsorbed on the crystal surfaces, and did not participate in the crystal structure.

### 3.5. Results of the Single-Crystal Diffraction 

In spite of several differences between α-HH and β-HH, their unit cells are recognized as identical due to their similar powder diffraction patterns [8]. The cell setting and crystal structure of these two forms of HH have been investigated for many years. Various experiments have been carried out to analyze the crystal structure, such as time-of-flight neutral powder diffraction [39], in situ time-resolved synchrotron radiation powder X-ray diffraction (SR-XRD) [40], X-ray powder diffraction [41], and single-crystal diffraction [8], which laid the foundation for us to study the lattice parameters and crystal structure of α-HH. The first structural model for α-HH was proposed by Gallitelli (1933) based on single-crystal X-ray data [42]. Later, using optical data, Flörke (1952) suggested that the structures of α-HH existed in both low- and high-temperature forms (trigonal above 318 K and orthorhombic below this temperature) [43]. Bushuev (1982) subsequently determined the crystal structure of the compound CaSO4·0.67H_2_O from single-crystal X-ray intensity data [44,45]. Ballirano prepared the α-HH by the rehydration of γ-anhydrite and concluded that the space group was I2 (unique axis b) [41]. According to Table 3, it can be seen that the models proposed by different researchers differ in their crystal structure and cell parameters.

In order to prepare crystals with high crystallinity, we adjusted the growth conditions and performed hydrothermal experiments. Under the conditions of slow heating and low-speed stirring, crystals with a uniform size and minor twinning were obtained (as shown in Figure 7). The results of the single-crystal diffraction experiment in this work were as follows: cell parameters *a* = 12.658(3) Å; *b* = 13.855(3) Å; *c* = 13.550(3) Å; β = 117.79(3)°; space group C2. The crystal structure is shown in Figure 8 (drawn in Diamond). The structure contained chains of alternating CaO_8_ and CaO_9_ coordination polyhedra held together by SO_4_^2−^ ions, which was in agreement with the conclusion in the literature [47]. The Ca-O bond distances in the CaO_8_ coordination polyhedral ranged from 0.2362 to 0.2588 nm, whereas in the CaO_9_ coordination polyhedral, it ranged from 0.2384 to 0.2688 nm. The average Ca-O distances were 0.2456 and 0.2515 nm, respectively. It was clear that the arrangement of the atoms was consistent with that in the other literature, but the values of the unit cell parameters were slightly different. Due to the hexagonal platelet shape (with an aspect ratio of lower than 0.5), the directions of the three axes chosen by the analyst differed from traditional hexagonal prism-shaped crystals (with an aspect ratio of higher than 1). As a consequence, the *a*-axis was identical to the *c*-axis in the other literature. In the following discussion, the experimental data of the *a*-axis and *c*-axis were exchanged, in line with the data of other researchers. Hence, the cell parameters from the single-crystal diffraction test were modified as follows: *a* = 13.550(3) Å; *b* = 13.855(3) Å; *c* = 12.658(3) Å; β = 117.79(3)°; space group C2, and the coordinates of the atoms are shown in Table 4. In this study, the values of the *a*-axis and *b*-axis were approximately twice of the values found by some other authors (Table 3), which might have been caused by different methods of crystal cells selected by analysts. Additionally, the value of β was higher than that in most of the literature, which was approximately 90°. Only Weiss and Bräu (2009) [53] and Schmidt et al. (2011) [8] suggested that the value of β was approximately 133° through the data of the single crystal test and refined program. In addition, the analysts did not find the elements of Al and C inside or on the edge of the structure during the process of sample analysis, which indicated that these elements were amorphous, were merely adsorbed on the surface of the crystal and did not participate in constructing the structure.

### 3.6. Analysis of the α-HH Crystal Structure 

Atoms in a crystal are arranged periodically, and the smallest component of the crystal is called the unit cell. One can partially predict crystal growth and external forms according to the crystal structure. In order to reveal the growth mechanism of the α-HH crystal, the distance between atoms was measured, and the atoms on different crystal facets were analyzed according to the unit cell structure. 

The distance between Ca atoms and the center of SO_4_ tetrahedra was simplified as the distance between Ca and S atoms. As illustrated in Figure 9, the distance was approximately 3.68 Å along the *a*-and *b*-axis, while it was approximately 3.16 Å along the *c*-axis. Therefore, the bond between Ca and SO_4_ tetrahedra along the *c*-axis was stronger than that in the other two directions.

In the XRD pattern of the α-HH crystals with a columnar shape (Figure 10), the 2θ of the characteristic diffraction peaks were 14.77°, 25.69°, 29.76° and 31.94°. According to the PDF (no.41-0024) information, the corresponding crystal faces were (200), (020), (400), and (204), respectively. The atoms on these faces were examined by building one 2 × 2 × 2 super cell and cutting it through the operation of “create lattice planes” in Diamond. The results are shown in Figure 11. They indicate that the (204) plane, as one of the end surfaces, is mainly composed of Ca^2+^, wheras the (400) and (200) planes consist of Ca^2+^ and SO_4_^2−^, and the (020) plane consists of O atoms. The (200), (400) and (020) planes are parallel to the *c*-axis, as side planes. It can be concluded that the positive charge in the (204) plane is larger than that of the other planes.

## 4. Discussion

The growth mechanism of α-HH crystals has been investigated for many years. However, no consensus has yet been reached by authors. Generally speaking, there are three views accepted by most scholars [10], and they are as follows: One is described as a topotactic solid state reaction in initial phase, then as a dissolution–recrystallization mechanism in the later phase. The second one assumes that the DH is first decomposed into AH and free water, and then the AH combines with water to form HH crystals. The third one is the dissolution–recrystallization mechanism, which is the dissolution of DH and generation of a supersaturation with respect to HH followed by the nucleation and growth of nuclei to macroscopic crystals from solution or on the surface of mother gypsum crystals without directional correlation [54]. 

The results of XRD and Section 3.3 were contrary to the second view of the growth mechanism above that there was no intermediate phase during the transformation from DH to α-HH, as proven in the literature using the method of in situ time-resolved synchrotron radiation powder X-ray diffraction [40]. However, this requires further investigation to determine whether the growth mechanism is a topotactic solid state reaction followed by dissolution–recrystallization or just a dissolution–recrystallization mechanism. Currently, the dissolution–recrystallization mechanism is more acceptable.

As shown in Figure 4b–d, it could be seen that the crystal growth finished first along the *a*-axis and *b*-axis. With the addition of an appropriate modifier, the crystal along the *c*-axis was divided into many fiber strips due to the inhibition of the modifier, and the fiber strips overlapped with each other to form a complete crystal with a short column shape as the reaction progressed. However, without the addition of a modifier, the crystal would grow preferentially along the *c*-axis into a needle shape with high aspect ratio. This could be explained by the results of the single-crystal experiment and the Hartman–Perdok theory. According to the Hartman–Perdok theory [55], there are a series of uninterrupted chains formed by strong bonds in the crystal, and the fastest direction of crystal growth is the direction with the strongest chemical bond. In the crystal structure of α-HH, if we ignore the effects of the water molecules in the channel, the crystal growth is mainly caused by the package of the alternating growth units Ca and SO_4_, and the growth mechanism can be revealed by measuring and comparing the distance between Ca and S atoms. As described in Section 3.6, the bond length of the Ca -SO_4_ chain along the *c*-axis was the shortest, and the bond was the strongest. As a result, if there is no other interference from external factors, the crystal will preferentially grow along the c-axis, which is consistent with the phenomenon observed from the reaction in pure water medium [26], and the crystal will develop into a fibrous shape.

However, the morphology of the α-HH crystal will vary in the presence of different crystal modifiers. According to the analysis of the atoms on different planes, the (204) plane was mainly composed of Ca ions with more positive charges, which tended to absorb carboxylate groups with negative charges. The carboxylate groups complexed Ca^2+^ ions to form a network, which prevented growth unit SO_4_ from adhering to the crystal face. As a consequence, the growth along the *c*-axis was inhibited, and the relative growth rates along the *a*- and *b*-axis increased, resulting in a columnar shape. As the reaction proceeded, the network formed by the carboxylate group and Ca ions was destroyed under the driving force of temperature and supersaturation, and the top surface developed into a flat plane, indicating that the growth along the *c*-axis had been completed (Figure 4d). These results are similar to the conclusions in [10] and [18].

In addition, HH was slightly soluble in aqueous medium. With the addition of a small amount of soluble aluminum sulfate into the reaction system, the concentration of SO_4_^2−^ increased, accompanied by an increase in supersaturation and driving force. Then, the nucleation rate accelerated, whereas the size of the crystals decreased, as demonstrated by Mi Y et al. [56] and Run Yang et al. [57]. Therefore, the amount of aluminum sulfate must be appropriate.

## 5. Conclusions

Based on the experiments mentioned above, it was proven that α-HH could be produced from FGD gypsum using potassium sodium tartrate and aluminum sulfate as the crystal modifier. In the dynamic process, the phase transformation started at approximately 140 °C and finished after being left at 150 °C for 30 min.

The FTIR analysis showed that the α-HH samples were accompanied by a small amount of organic substance due to the asymmetric and symmetric stretching vibrations of methylene (−CH_2_−) and suggested that a mutual reaction might exist between potassium sodium tartrate and the α-HH crystal since the intensity of the characteristic peak enhanced with the increase in the dosage. In order to gain further evidence on the mutual reaction, DH with fewer impurities should be used in the experiment.

The single-crystal diffraction test showed the arrangements of the atoms and the structure of the α-HH. The results of the lattice parameters were as follows: *a* = 13.550(3); *b* = 13.855(3); *c* = 12.658(3); β = 117.79(3)°; space group C2. The growth mechanism of α-HH crystals was revealed by measuring the bond length in the structure and analyzing the atom composition of the exposure crystal surfaces. This work provided a new insight into the structure of the unit cell. However, more perfect α-HH crystals need to be prepared for single-crystal diffraction in order to obtain more precise cell parameters, and the growth mechanism considering the role of the water molecules in the channel should be further studied. At the same time, the process of α-HH production needs be improved to reduce energy consumption and increase production efficiency.

## Figures and Tables

**Figure 1 materials-15-03382-f001:**
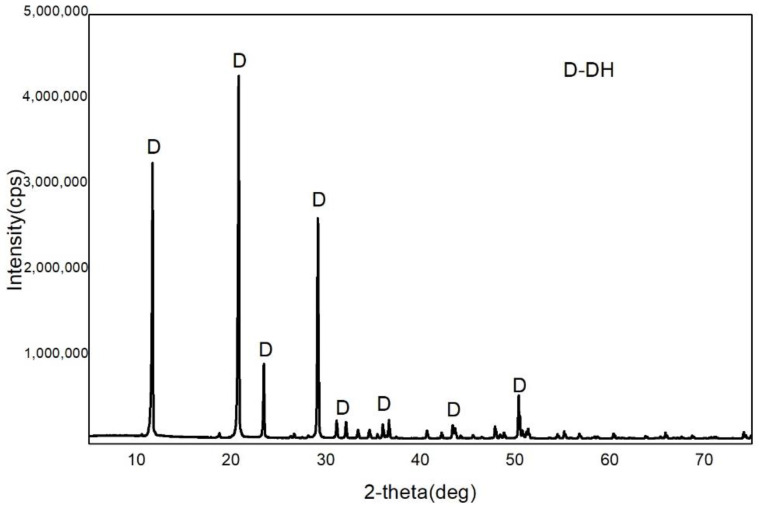
X-ray diffraction (XRD) pattern of the FGD gypsum.

**Figure 2 materials-15-03382-f002:**
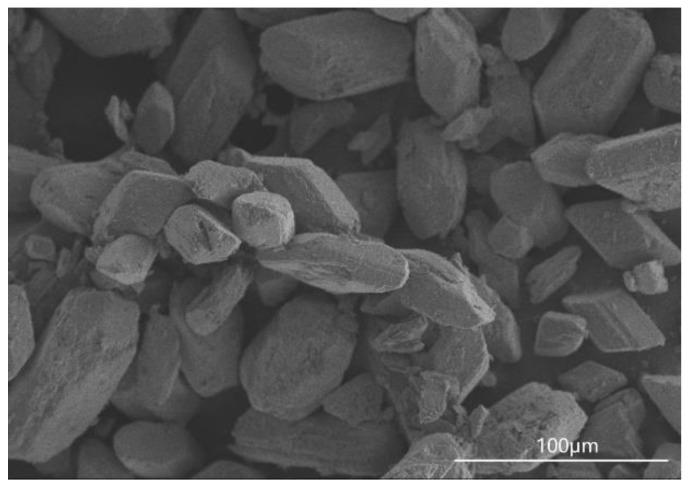
Morphology of the FGD gypsum crystals.

**Figure 3 materials-15-03382-f003:**
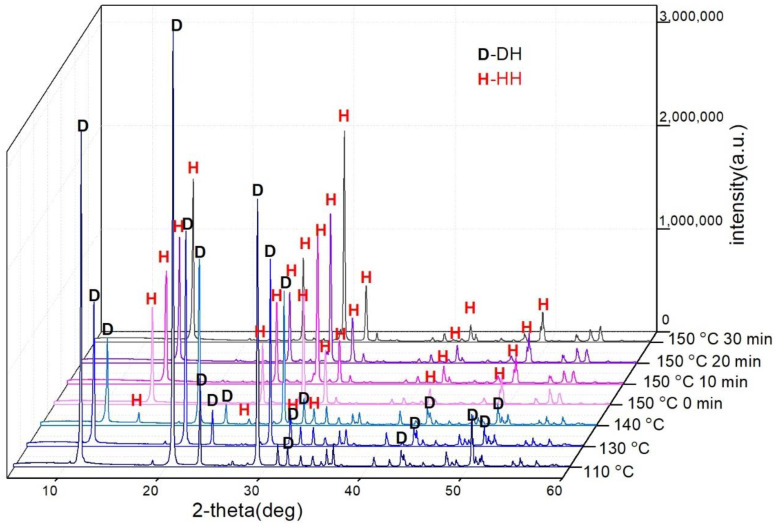
XRD patterns of the samples corresponding to the reaction stages of 110 °C, 130 °C, 140 °C, 150 °C, 150 °C lasting for 10 min and 150 °C lasting for 30 min.

**Figure 4 materials-15-03382-f004:**
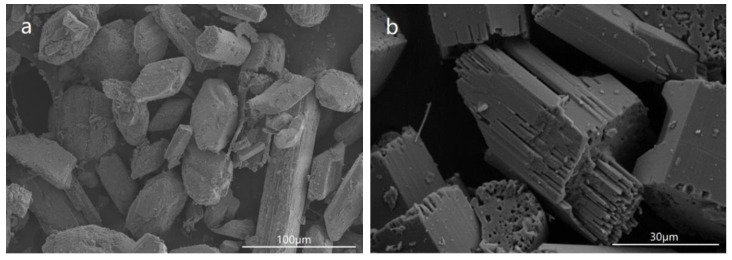
SEM images of samples withdrawn at the reaction stages of (**a**) 140 °C, (**b**) 150 °C (**c**) 150 °C lasting for 10 min and (**d**) 150 °C lasting for 30 min.

**Figure 5 materials-15-03382-f005:**
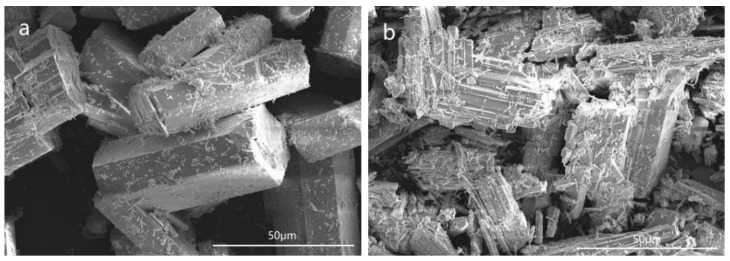
The morphology of hemihydrate gypsum prepared from dihydrate gypsum contained 15 wt% (**a**) and 0 (**b**) of attachment water.

**Figure 6 materials-15-03382-f006:**
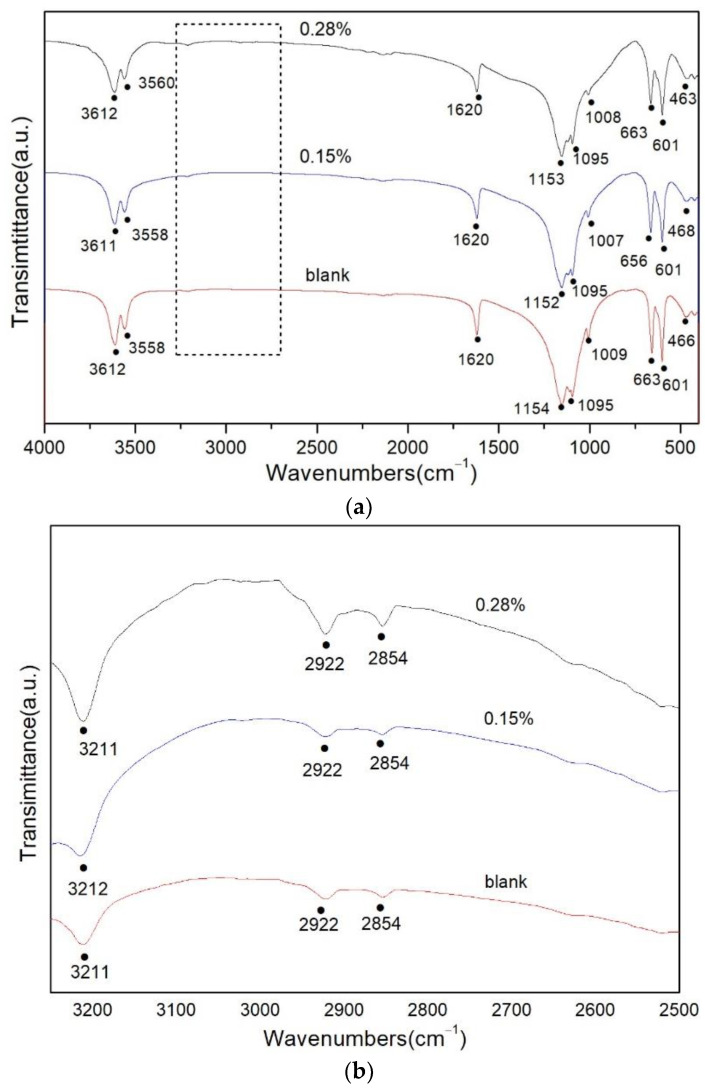
FTIR spectra of (**a**) samples obtained from different dosage of potassium sodium tartrate; (**b**) magnified picture (from 2500 cm^−1^ to 3250 cm^−1^ as the part of the dashed box in (**a**)).

**Figure 7 materials-15-03382-f007:**
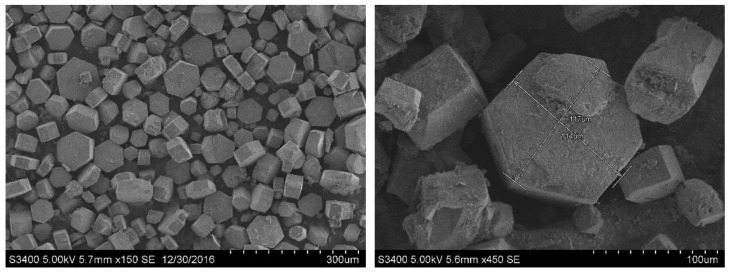
SEM images of the crystals prepared for single-crystal diffraction.

**Figure 8 materials-15-03382-f008:**
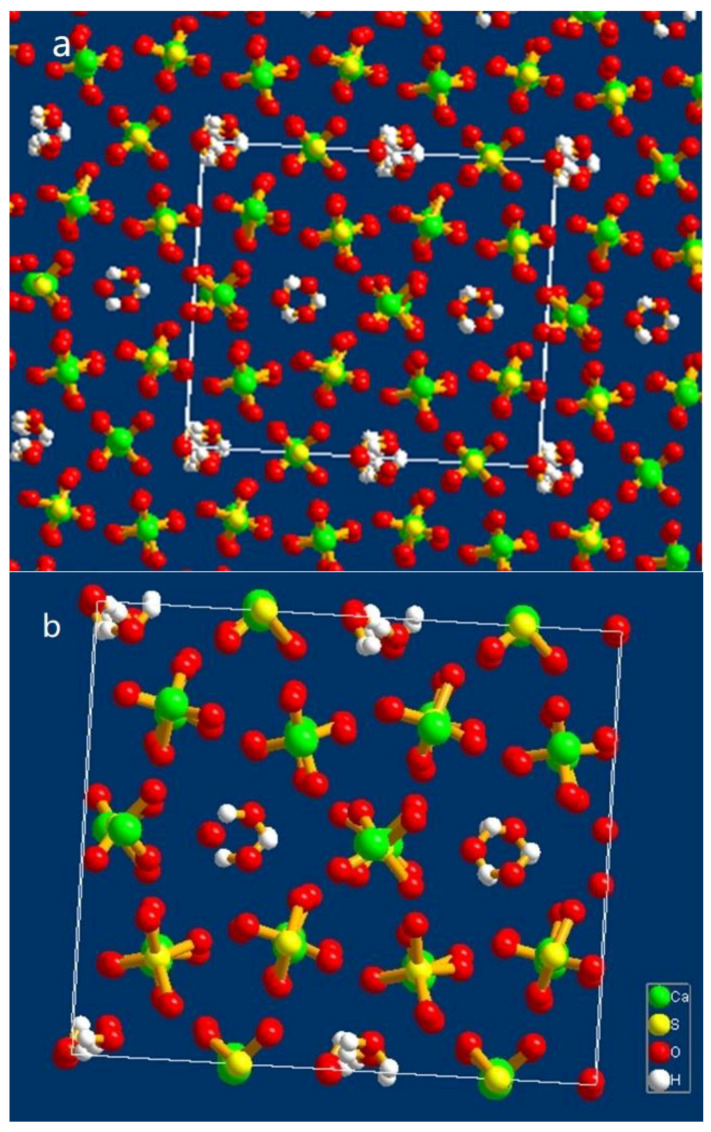
The crystal structure of α-HH along *c*-axis: (**a**) multiple cells; (**b**) unit cell. Here, the square box represents the unit cell edges.

**Figure 9 materials-15-03382-f009:**
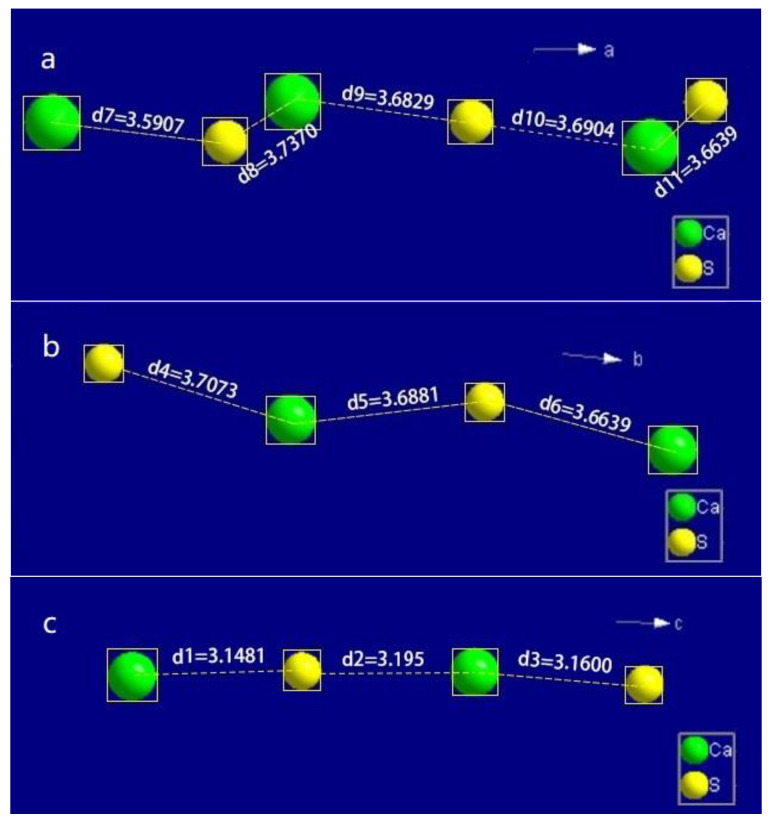
The distances between the Ca ions and the S atoms in the unit cell along (**a**) the *a*-axis, (**b**) the *b*-axis and (**c**) the *c*-axis.

**Figure 10 materials-15-03382-f010:**
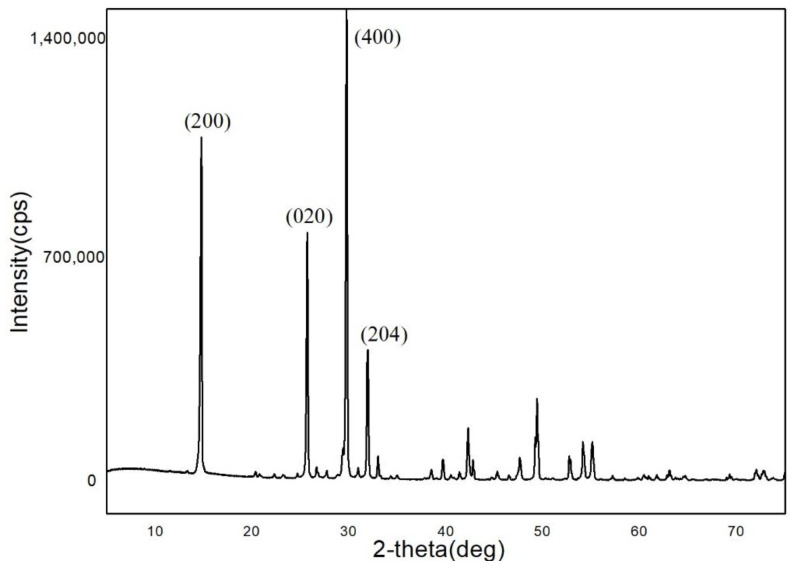
XRD pattern of the samples with columnar shape.

**Figure 11 materials-15-03382-f011:**
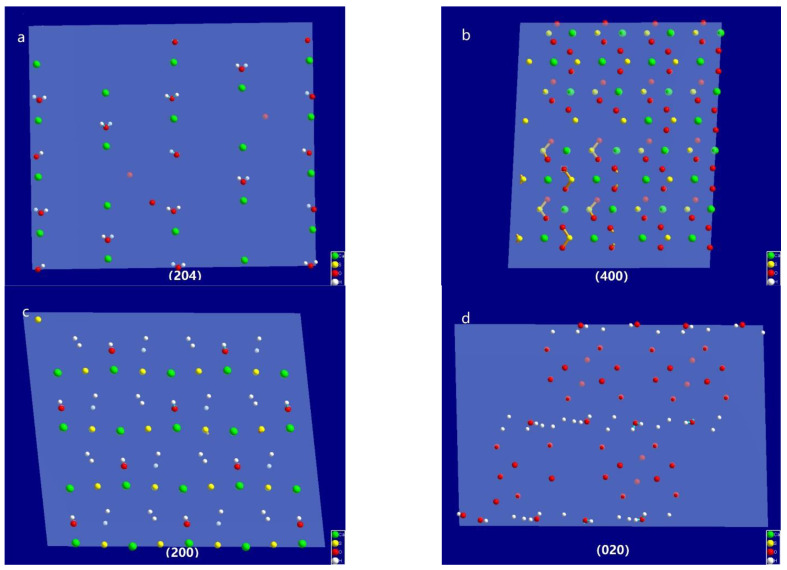
Atoms on the crystal plane of (**a**) (204), (**b**) (400), (**c**) (200) and (**d**) (020).

**Table 1 materials-15-03382-t001:** The chemical component of the Flue-gas desulphurization (FGD) gypsum (wt%).

SO_3_	CaO	H_2_O	SiO_2_	Al_2_O_3_	Fe_2_O_3_	K_2_O	MgO	Others
44.80	31.53	19.90	0.67	0.89	0.09	0.05	0.10	1.97

**Table 2 materials-15-03382-t002:** Physical performances of the α-hemihydrate gypsum (α-HH) plaster.

Items	Standard Index [27]	Results
Fineness (%)	≤5	4.0
Initial setting time(min)	≥3	15
Final setting time(min)	≤30	18
Flexural strength of 2 h (MPa)	≥6.0	7.5
Oven-dry compressive strength (MPa)	≥50	65.0

**Table 3 materials-15-03382-t003:** Crystallographic data for calcium sulfate hemihydrate (CaSO_4_·x H_2_O, 0.5 ≤ x ≤ 0.8) from the literature.

Reference	Space Group	*a*(Å)	*b(*Å)	*c*(Å)	*Β*(°)	H_2_O
Gallitelli (1933) [42]	C2	11.94	6.83	12.70	~90	0.5
	P3_1_21	6.83	6.83	12.70	120(γ)	0.5
Caspari (1936) [46]	P$\overset{–}{3}$m1	6.82	6.82	6.24		0.5
Flörke (1952) [43]	C222	6.83	11.49	12.70	90	0.5
	P3_2_21	6.83	6.83	12.70	90	0.5
Gay (1965b) [47]		6.85	11.88	12.60	~90	0.5
Frik and Kuzel (1982) [48]		12.061	6.933	12.670		0.48
		13.865	13.865	12.718		0.52
Bushuev and Borisov (1982) [45]	I2	12.028	12.674	6.927	90.21(γ)	0.67
	P3_1_21	6.977	6.977	12.617		0.5
Abriel (1983) [49]	P3_1_21	6.968	6.968	6.410		0.8
Lager (1984) [39]	I2	12.062	12.660	6.930	~90(γ)	0.5
Kuzel and Hauner (1987) [50]	I2	12.0275	6.9312	12.6919	90.18	0.5
	P3_1_21	13.8615	13.8615	12.7391		0.66
Abriel and Nesper (1993) [51]	I2	12.0275	6.9312	12.6919	90.18	0.53
Bezou et al. (1995) [52]	I2	12.0317	6.9272	12.6711	90.265	0.5
	I2	11.9845	6.9292	12.7505	90	0.6
Ballirano et al. (2001) [41]	I2	12.0350	6.9294	12.6705	90.266	0.5
Weiss and Bräu (2009) [53]	C2	17.559	6.9619	12.071	133.56	0.5
Schmidt et al. (2011) [8]	C2	17.5180	6.9291	12.0344	133.655	0.5
	P3_2_21	13.8690	13.8690	12.7181		0.625

**Table 4 materials-15-03382-t004:** Coordinates of CaSO_4_·0.5H_2_O from the single-crystal diffraction text.

Atom	x/*a*	y/*b*	z/*c*	B(Å^2^)
Ca1	0.7364(3)	0.4037(2)	1.2848(3)	0.0125(7)
Ca2	0.5000	0.0762(3)	1.000	0.0055(8)
Ca3	0.7837(2)	0.1520(2)	0.9738(3)	0.0113(6)
Ca4	0.5000	0.0485(3)	0.5000	0.0079(8)
Ca5	0.7189(3)	0.4027(2)	0.7769(3)	0.0125(7)
Ca6	0.9910(3)	0.3109(3)	0.7450(3)	0.0175(7)
Ca7	1.2374(3)	0.1541(2)	0.5361(3)	0.0146(7)
S1	0.7287(3)	0.3995(3)	1.0342(3)	0.0085(6)
S2	0.7778(3)	0.1527(3)	1.2232(3)	0.0129(7)
S3	0.5064(3)	0.0653(3)	0.7492(3)	0.0071(6)
S4	0.7723(3)	0.1508(3)	0.7187(3)	0.0119(7)
S5	1.0000	0.3181(5)	1.0000	0.0136(9)
S6	0.7220(3)	0.4044(3)	0.5246(3)	0.0092(6)
S7	1.0000	0.3112(5)	0.5000	0.0129(9)
O1	0.8220(8)	0.3862(8)	1.1491(9)	0.010(1)
O2	0.6330(8)	0.4178(7)	1.0606(9)	0.0093(10)
O3	0.7433(8)	0.4816(8)	0.9684(9)	0.0099(10)
O4	0.7064(8)	0.3113(8)	0.9576(8)	0.0091(10)
O5	0.8800(8)	0.1826(8)	1.2194(10)	0.0137(10)
O6	0.6941(8)	0.1300(8)	1.0987(9)	0.0136(10)
O7	0.7995(8)	0.0707(8)	1.3021(9)	0.0135(10)
O8	0.7361(9)	0.2301(8)	1.2719(10)	0.0134(10)
O9	0.4315(8)	0.0239(7)	0.7919(9)	0.0077(10)
O10	0.5873(7)	0.1265(7)	0.8431(8)	0.0074(9)
O11	0.4406(8)	0.1245(7)	0.6415(8)	0.0083(9)
O12	0.5609(8)	-0.0104(7)	0.7120(8)	0.0069(9)
O13	0.7974(8)	0.0645(8)	0.7931(9)	0.0126(10)
O14	0.6722(8)	0.1387(8)	0.6040(9)	0.0122(10)
O15	0.8643(8)	0.1707(8)	0.6912(9)	0.0131(10)
O16	0.7613(8)	0.2294(8)	0.7906(9)	0.0125(10)
O17	1.0561(9)	0.2554(9)	0.9530(9)	0.0143(12)
O18	0.9235(8)	0.3772(9)	0.8977(10)	0.0151(12)
O19	0.8075(8)	0.3808(8)	0.6445(9)	0.0107(10)
O20	0.7611(8)	0.4807(8)	0.4721(9)	0.0092(10)
O21	0.6964(8)	0.3194(8)	0.4433(8)	0.0096(10)
O22	0.6230(8)	0.4353(8)	0.5338(9)	0.010(1)
O23	1.0721(8)	0.2534(9)	0.4681(10)	0.0138(11)
O24	1.0689(8)	0.3727(8)	0.6071(9)	0.0139(11)
O25	0.5000	0.2426(19)	1.0000	0.048(6)
H25	0.5553	0.2755	1.0085	0.05800
O26	0.5485(11)	0.3174(11)	0.6886(12)	0.030(3)
H26A	0.5493	0.2700	0.7304	0.03600
H26B	0.4961	0.3513	0.6863	0.03600
O27	0.993(3)	0.491(3)	0.765(3)	0.045(7)
H27A	0.9928	0.5227	0.7087	0.05300
H27B	0.9353	0.5144	0.7640	0.05300
O29	0.039(4)	0.572(3)	0.424(5)	0.085(13)
H29A	0.0223	0.5435	0.3592	0.10200
H29B	0.0360	0.5254	0.4644	0.10200
O28	0.062(2)	0.568(2)	0.953(3)	0.039(7)
H28A	0.0626	0.5255	0.9998	0.04700
H28B	0.0099	0.6044	0.9462	0.04700

## Data Availability

Not applicable.

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
