# Peer review of "Experimental Study and Mechanism Analysis of Preparation of α-Calcium Sulfate Hemihydrate from FGD Gypsum with Dynamic Method"

_materials, 2022, doi:10.3390/ma15093382_

Round 1

Reviewer 1 Report

Author must needs to remove the  plagiarism and it should be below 10%.

The attached manuscript is marked with the comments for the revision. Though, the manuscript is facing high plagiarism issues. Hence, I was suggested to remove the plagiarism for smooth review.

Author Response

Thank you so much for your comments. Based on your comments, the author has made the following revisions:

  1. English language and style.

The author has already checked the manuscript carefully and revised some language questions. Due to lots of the revisions, the author will resubmit the manuscript again and apply for the editing services of MDPI.

  1. Introduction, must be improved.

The author has rewritten the introduction again, and there may be other problems.  Hope your valuable comments again. (Please see the section 1)

  1. Research design, can be improved.

The author has analyzed the experiment results in another way to make it more reasonable. (Please see the section 3.6 and 4)

  1. Methods described, can be improved.

The author has described the method again in one more detailed way. (Please see the section 2)

  1. Results presented, can be improved.

The author has presented some of results more clearly. (Please see the section 3.6)

  1. Conclusions, can be improved.

The author has rewritten some of the conclusions and analyzed the results again to support the conclusions. (Please see the section 3.6 and 4)

  1. Plagiarism.

The author has rewritten most of the contents marked. (Please see the attachment)

Finally, we would like to thank you for the detailed and insightful comments again!

Sincerely with our best wishes!

Reviewer 2 Report

In the experiment section: Please describe the PH value of starting gypsum mixture for synthesizing the alpha hemihydrates. 

With respect to reviewing the paper titled "Experimental study and mechanism analysis of preparation of α-calcium sulfate hemihydrate from FGD gypsum with dynamic method", the paper discussed the preparation of alpha hemihydrate in aqueous media via saturated steam pressure (in an autoclave) with using crystal modifiers. The results were well presented with appropriate discussions. With respect to me, the conclusion was familiar, in which the effect of crystal modifier was clear on the morphology of alpha hemihydrate. 

Author Response

Thank you so much for your comments. Based on your comments, the author has made the following revisions:

  1. English language and style.

The author has already checked the manuscript carefully and revised some language questions. Due to lots of the revisions, the author will resubmit the manuscript again and apply for the editing services of MDPI.

  1. Methods described, can be improved.

The author has described the method again in one more detailed way. (Please see the section 2)

  1. Results presented, can be improved.

The author has presented some of results more clearly. (Please see the section 3.6)

  1. Please describe the PH value of starting gypsum mixture for synthesizing the alpha hemihydrates.

The author has already described the pH value of the mixture in section 2.2. (Please see the section 2.2)

Please see the attachment of revised manuscript.

Finally, we would like to thank you for the detailed and insightful comments again!

Sincerely with our best wishes!

Author Response

Thank you so much for your comments. Based on your comments, the author has made the following revisions:

  1. English language and style.

The author has already checked the manuscript carefully and revised some language questions. Due to lots of the revisions, the author will resubmit the manuscript again and apply for the editing services of MDPI.

  1. Introduction, can be improved.

The author has rewritten the introduction again, and there may be other problems.  Hope your valuable comments again.

  1. There have been many research results regarding this topic in recent years, which provide a theoretical basis for the understanding, in-depth research and application research of some system problems.

The author is well aware of her lack of theoretical knowledge and needs to study further. Thank you so much.

  1. The scientific value of the concerns of the application project and its potential contribution to relevant frontier fields, the proposed structures are widely used in actual engineering and practice.

The author has mentioned the actual application for FGD gypsum, phosphogypsum and citric acid gypsum in section 1.

  1. The assumption and limitation of this work should be clarified.

The author has presented the assumption and limitation in section3.6 and 4.

  1. The results of this paper should be validated with other published works.

The author has analyzed the results again to support the conclusions, and validated the results with some published works. (Please see section 4)

  1. The explanation of key scientific issues is not clear.

The author has analyzed the experiment results in another way to make it more reasonable. And the explanation of the growth mechanism has been rewritten again. (Please see the section 3.6 and 4)

Please see the attachment of revised manuscript.

Finally, we would like to thank you for the detailed and insightful comments again!

Sincerely with our best wishes!

Reviewer 4 Report

I have reviewed the manuscript “Experimental study and mechanism analysis of preparation of α-calcium sulfate hemihydrate from FGD gypsum with dynamic method” submitted to “Materials” for publication. In this study, authors have investigated a dynamic method with an improved autoclaved processing to transform FGD gypsum into α-calcium sulfate hemihydrate. This manuscript fits well within the scope of the journal; it needs some improvements; there are a few suggestions that authors may consider to improve it further:

The use of English language is reasonable, however, there are a number of punctuation and grammatical errors; that should be corrected and rephrased using academic English for a better flow of text for reader.

- Abstract is appropriate, however please add some more details about the methodology that is missing in the abstract section.

- The introduction seems well-structured and providing all the background information to the reader.

Figure 1 resolution is poor, will it be possible to improve?

Figure 2: lamellar morphology and related description should be added to the figure captain

Scalebar on all SEM images should have a similar format in all images

If possible, figures 8-11 can be condensed as 8a, 8b etc to reduce total number of figures.

Also legend showing S, H, O, and Ca is not present in all the images; please check

Please discuss, why SEM was chosen for the characterization of morphology?/preferred over other techniques? Why the elemental analysis was not performed?

- Please elaborate on the limitations and the future research directions in the discussion section.

Author Response

Thank you so much for your comments. Based on your comments, the author has made the following revisions:

  1. English language and style. The use of English language is reasonable, however, there are a number of punctuation and grammatical errors; that should be corrected and rephrased using academic English for a better flow of text for reader.

The author has already checked the manuscript carefully and revised some language questions. Due to lots of the revisions, the author will resubmit the manuscript again and apply for the editing services of MDPI.

  1. Research design, can be improved.

The author has analyzed the experiment results in another way to make it more reasonable. (Please see the section 3.6 and 4).

  1. Results presented, can be improved.

The author has presented some of results more clearly. (Please see the section 3.6)

  1. Abstract is appropriate, however please add some more details about the methodology that is missing in the abstract section.

The author has added a description of the method into the abstract. (Please see Abrstract.)

  1. The introduction seems well-structured and providing all the background information to the reader.

The author has revised some contents of the introduction again. (Please see the section 1)

  1. Figure 1 resolution is poor, will it be possible to improve?

The author has drawn the picture again. (Please see Figure 1)

  1. Figure 2: lamellar morphology and related description should be added to the figure captain.

The author has described Figure 2 again as the thick plate shape of crystals in the section 2.1. (Please see section 2.1)

  1. Scalebar on all SEM images should have a similar format in all images.

The author has changed the picture of Figure 2 with a similar format. (Please see the Figure 2)

  1. If possible, figures 8-11 can be condensed as 8a, 8b etc to reduce total number of figures..

The author has condensed Figure 8 and 9 as Figure 8a and 8b. (Please see the Figure 8)

  1. Also legend showing S, H, O, and Ca is not present in all the images; please check.

The author has checked all the images and put on the legend showing. (Please see the Figure 8,9 and 11)

  1. Please discuss, why SEM was chosen for the characterization of morphology?/preferred over other techniques? Why the elemental analysis was not performed?

SEM is more convenient for the author, not more advanced than other techniques. The author is inexperienced. It is the author's fault that the element analysis is not carried out. The author will strengthen the cultivation of experimental skills in future study. I am so grateful to you for this point.

  1. Please elaborate on the limitations and the future research directions in the discussion section.

The author has described the limitations in the section 4 and the research directions in section 5. (Please see the section 4 and 5)

Please see the attachment of revised manuscript.

Finally, we would like to thank you for the detailed and insightful comments again!

Sincerely with our best wishes!

Round 2

Reviewer 1 Report

Author has attended all the comments. Good luck!

Reviewer 3 Report

This paper can be accepted for publication

Reviewer 4 Report

Many thanks for revising the manuscript.